# ERTFM: An Effective Model to Fuse Chinese GF-1 and MODIS Reflectance Data for Terrestrial Latent Heat Flux Estimation

**Lilin Zhang** [1,2], **Yunjun Yao** [1,*], **Xiangyi Bei** [1], **Yufu Li** [3], **Ke Shang** [1], **Junming Yang** [1], **Xiaozheng Guo** [1], **Ruiyang Yu** [1] **and Zijing Xie** [1]

1 State Key Laboratory of Remote Sensing Science, Faculty of Geographical Science, Beijing Normal University, Beijing 100875, China; l.zhang-2@utwente.nl (L.Z.); xiangyibei@mail.bnu.edu.cn (X.B.); shangke@mail.bnu.edu.cn (K.S.); julming@mail.bnu.edu.cn (J.Y.); boyxiaozheng@mail.bnu.edu.cn (X.G.); yuruiyang@mail.bnu.edu.cn (R.Y.); xiezijing@mail.bnu.edu.cn (Z.X.)
2 Faculty of Geo-Information and Earth Observation (ITC), University of Twente, 7500 Enschede, AE, The Netherlands
3 Jincheng Meteorological Administration, Jincheng 048026, China; qxtlyf@163.com
* Correspondence: yaoyunjun@bnu.edu.cn

**Abstract:** Coarse spatial resolution sensors play a major role in capturing temporal variation, as satellite images that capture fine spatial scales have a relatively long revisit cycle. The trade-off between the revisit cycle and spatial resolution hinders the access of terrestrial latent heat flux (LE) data with both fine spatial and temporal resolution. In this paper, we firstly investigated the capability of an Extremely Randomized Trees Fusion Model (ERTFM) to reconstruct high spatiotemporal resolution reflectance data from a fusion of the Chinese GaoFen-1 (GF-1) and the Moderate Resolution Imaging Spectroradiometer (MODIS) products. Then, based on the merged reflectance data, we used a Modified-Satellite Priestley–Taylor (MS–PT) algorithm to generate LE products at high spatial and temporal resolutions. Our results illustrated that the ERTFM-based reflectance estimates showed close similarity with observed GF-1 images and the predicted NDVI agreed well with observed NDVI at two corresponding dates (r = 0.76 and 0.86, respectively). In comparison with other four fusion methods, including the widely used spatial and temporal adaptive reflectance fusion model (STARFM) and the enhanced STARFM, ERTFM had the best performance in terms of predicting reflectance (SSIM = 0.91; r = 0.77). Further analysis revealed that LE estimates using ERTFM-based data presented more detailed spatiotemporal characteristics and provided close agreement with site-level LE observations, with an $R^2$ of 0.81 and an RMSE of 19.18 W/$m^2$. Our findings suggest that the ERTFM can be used to improve LE estimation with high frequency and high spatial resolution, meaning that it has great potential to support agricultural monitoring and irrigation management.

**Keywords:** Extremely Randomized Trees; latent heat flux; high spatial and temporal resolutions; Chinese GF-1; MODIS

## 1. Introduction

Latent heat flux (LE) refers to the heat flux transferred to the atmosphere in the process of surface soil evaporation, vegetation transpiration, and canopy intercepted evaporation, and is an important component in water balance and the energy cycle [1–3]. Spatiotemporally continuous LE is of considerable significance to a variety of studies including understanding water and energy exchange [4], renewing terrestrial freshwater resources [5] and climate change forecasting [6]. The accurate estimation of LE with both high frequency and high spatial resolution is urgently desired for regional water resources management and irrigation decision making in agriculture.

Satellite-based observations provide an unprecedented opportunity to monitor large-area terrestrial ecosystem dynamics, and have been used extensively in land surface-related estimates and applications from regional to global scales [7,8]. Over the last few decades,

a large number of satellite-derived LE products with different spatial resolutions have been generated for terrestrial monitoring applications. To our knowledge, available LE products have either high spatial resolution or frequent revisit cycle, such as the Moderate Resolution Imaging Spectroradiometer (MODIS) LE product (1 km, 8 days) [9,10], the Global Land Evaporation Amsterdam Model (GLEAM) LE product (0.25°, daily), and the Mapping EvapoTranspiration with Internalized Calibration (METRIC) LE product (30 m, 16 days) [11]. It is still a challenge to obtain remotely-sensed LE with both high spatial resolution and dense acquisition frequency, due to the trade-off between revisit cycles and spatial resolution.

Currently, the MODIS LE product has been adopted as one of the most widely used data sources, with daily acquired coverage and moderate spatial resolutions ranging from 250 m to 8 km. However, due to the lack of detailed land surface information, its relatively coarse spatial resolution is not sufficient to characterize the variations of LE in heterogeneous areas [12]. With the emergence of new satellite sensors, such as the Chinese GaoFen-1 Wide Field View (GF-1 WFV) (16 m spatial resolution and frequent revisit cycle) and Sentinel-2 (a five-day cycle and 10 m spatial resolution), it has been possible to provide highly valuable data sources at fine spatial resolution [13,14]. Nevertheless, cloud contamination seriously interferes with image acquisition, contributing to a generally sparse temporal frequency. Therefore, there is an urgent need to combine the merits of the two types of satellite images and develop the spatiotemporal fusion method [15].

In recent years, many efforts have been devoted to addressing this issue and the work that has been done has shown that it is possible for spatiotemporal fusion methods to generate imagery with high spatial and temporal resolution [16,17]. According to the principles of the different techniques, the current fusion methods can be classified into five categories: weight function-based, unmixing-based, Bayesian-based, learning-based and hybrid methods [18]. Table 1 lists some typical spatiotemporal fusion methods. All of these methods promise enhancements in spatial and temporal resolution and each has its own unique advantages when it comes to taking different types of input data, levels of computational efficiency and application requirements.

**Table 1.** Typical spatiotemporal fusion methods and related experimental data.

| Category | Description | Method | References | Experimental Data |
|---|---|---|---|---|
| Weight function-based | Introduced adjacent similarity pixel information to predict the target pixels and combine spectral similarity, spatial distance, as well as temporal differences | STARFM | Gao et al. [19] | MODIS, Landsat |
| | | ESTARFM | Zhu et al. [20] | MODIS, Landsat |
| | | STAARCH | Hilker et al. [21] | MODIS, Landsat |
| | | Semi-Physical Fusion Approach | Roy et al. [22] | MODIS, Landsat |
| | | SADFAT | Weng et al. [5] | MODIS, Landsat |
| | | RWSTFM | Wang et al. [23] | MODIS, Landsat |
| Unmixing-based | Definition of endmembers, unmixing of coarse pixels, and assignment of pixels to fine classes | MMT | Zhukov et al. [24] | Landsat, MERIS |
| | | Constrained unmixing | Zurita-Milla et al. [25] | Landsat, MERIS |
| | | LAC-GAS | Maselli et al. [26] | AVHRR LAC, GAC NDVI |
| | | STDFA | Wu et al. [27] | MODIS, Landsat |
| Bayesian-based | Based on the Bayesian theory, developed the maximum posterior probability model to estimate the fine pixel value | BME | Li et al. [28] | MODIS, AMSR-E |
| | | Spatio-Temporal-Spectral fusion | Xue et al. [29] | MODIS, Landsat |
| Learning-based | Adopted machine learning to establish correspondences between fine and coarse datasets | SPSTFM | Huang et al. [30] | MODIS, Landsat |
| | | ELM learning | Liu et al. [31] | MODIS, Landsat |
| | | Fit-FC | Wang et al. [32] | Sentinel-2, Sentinel-3 |
| | | MRT | Xu et al. [33] | MODIS, Landsat |
| | | ESRCNN | Shao et al. [34] | Landsat, Sentinel-2 |
| Hybrid methods | Combined the advantages of two or more of the above four methods to improve fusion performance | FSDAF | Zhu et al. [35] | MODIS, Landsat |
| | | STRUM | Gevaert et al. [35] | MODIS, Landsat |
| | | STIMFM | Li et al. [36] | MODIS, Landsat |

Among these mainstream fusion methods, the spatial and temporal adaptive reflectance fusion model (STARFM) proposed by Gao et al. [19] has been widely used. STARFM introduces the information of spectral similarity between pixels and further utilizes a weight function to characterize the contribution of each neighboring pixel. Semmens et al. [37], using STARFM, estimated the daily latent heat flux with 30 m resolution at irrigated agricultural fields and achieved a relative mean absolute of approximately 19–23%. To improve the performance of the fusion algorithm, STARFM was further modified and upgraded for various complex situations. For instance, STAARCH, developed by Hilker et al. [21], can effectively capture terrestrial disturbance and monitor landcover change. The enhanced STARFM (ESTARFM), proposed by Zhu et al. [20], adds the conversion coefficient to address the spatial information, which is more suitable for highly heterogeneous areas. In addition, based on the unmixing theory, a number of spatiotemporal fusion approaches have been developed, such as the multisensor multiresolution technique (MMT) [24], STDFA [27]. However, these approaches are for the most part based on the assumption that the relationship between known and predicted pixels is one of linear change. Due to the complexity and variability of land surface, a large number of nonlinear mixed pixels exist in satellite images of different phase, resulting in a decrease in the accuracy of prediction. Thus, there is scope for improving the accuracy of spatiotemporal fusion methods over areas with heterogeneous landscapes.

Inspired by the excellent nonlinear representation ability of machine learning, learning-based fusion techniques have the potential to improve the accuracy of prediction in heterogenous landscapes. Several learning-based spatiotemporal fusion methods have recently been developed. For instance, the Sparse-representation-based Spatiotemporal reflectance Fusion Model (SPSTFM), put forward by Huang and Song [30], learned the corresponding relationship between the available MODIS–Landsat image pairs, and then was applied to predict the fine image with a high level of accuracy. In addition, the Extreme Learning Machine (ELM) fusion model [30], random forest (RF) [38] machine learning and the convolutional neural network (CNN)-based fusion network [34] have also attracted attention in the attempts to reconstruct fine images. Ke et al. [38] designed the RF-based downscaling approaches that combine RF and spatiotemporal fusion algorithms to generate an 8-day LE product at a spatial resolution of 30 m, which provides a unique approach for monitoring water and energy exchange at regional scales.

Extremely Randomized Trees (ERT), as a novel decision tree-based learning method, has performed well in downscaling [39]. Compared with RF, the ERT model has stronger generalization and its node splitting is generally more random [40]. As the number of decision trees increases, the error rate is expected to significantly decrease [41]. To date, several researches have confirmed that the regression accuracy of the ERT algorithm is superior to RF and ANN [42,43]. However, to our knowledge, the ERT algorithm has not been adopted to reconstruct the spatiotemporally continuous LE product. Given that the spatiotemporal fusion model based on the ERT algorithm can be easily developed from the available image pairs without complex formulas, we tried to combine it with a simple but solid satellite-derived LE model, namely, the Modified-Satellite Priestley–Taylor (MS–PT) model developed by Yao et al. [44], to construct an LE product with high spatial and temporal resolutions. Departing from the traditional machine learning-based downscaling framework, we took advantage of the powerful ability of ERT nonlinear fitting to learn a mapping function based upon the MODIS–GF-1 image pairs difference.

In this paper, we focus on the implementation of an Extremely Randomized Trees Fusion Model (ERTFM) along with the MS–PT model to generate a spatiotemporally continuous LE product at 16 m daily resolution. We had three major objectives: First, we proposed the ERTFM model to reconstruct high spatiotemporal resolution reflectance data from a fusion of GF-1 and MODIS products and then assessed its accuracy with observed GF-1 images. Second, we compared the ability of ERTFM for downscaling to four other typical fusion methods. Third, we applied the ERTFM and the MS–PT model to generate a high spatiotemporal LE product and then compared it with in situ LE observations.

## 2. Materials and Methods

### 2.1. The ERTFM Logic

Extremely Randomized Trees (ERT) is a nonlinear and nonparametric machine learning model designed by Geurts et al. [39] to address classification and regression problems. The ERT algorithm established an ensemble decision or regression tree according to the classical top-down procedure, similar to standard tree-based ensemble methods. Compared with other tree-based methods, e.g., RF, the ERT splits nodes totally randomly and utilizes the whole learning sample to build the tree nodes rather than bootstrapping replicas [40]. In addition, the algorithm selects a cut-point or attributes for each feature completely randomly, instead of calculating the locally optimal one. In terms of accuracy and computational efficiency, ERT achieved a satisfactory performance.

There are three key parameters in the procedures, namely K, $N_{min}$ and M, where K denotes the number of splitting nodes, $N_{min}$ denotes the number of samples of cut-point, and M denotes the number of regression trees [39]. The ERT model will automatically adjust their values to get the optimal solution, according to the learning samples. The ERT has shown powerful generalization and robustness, and can partly address the overfitting problems of other machine learning through using more random splits nodes, which is more likely to keep surface variances over a heterogeneous landscape [39,41]. It is worth noting that Shang et al. [43] succeeded in applying the ERT approach to integrate the five LE products from across Europe. Therefore, the ERT has the enormous potential to derive an LE product with a high spatiotemporal resolution.

In contrast to the sophisticated learning-based fusion method, we were mainly concerned with feature mapping from the coarse-fine image pairs, and thereby avoided the complexity of sparse coding. The ERT approach was employed to train the mapping function. In this study, two pairs of coarse-fine images at $T_1$ and $T_3$ and an extra coarse image at $T_2$ were acquired to predict a fine resolution image. GF-1 ($G_1$, $G_3$) and MODIS ($M_1$, $M_3$) denotes the reflectance images at the date of $T_1$ and $T_3$. We proposed the ERT fusion model (ERTFM) based on the MODIS and GF-1 data to generate the high spatiotemporal resolution LE product. The flowchart of the ERTFM is shown in Figure 1 and the schematic overview of ERTFM is described below.

Firstly, Training Samples Generation. The ERT was adopted to train the mapping function between the GF-1 and MODIS images. The different images were directly used for model construction, with 70% of the samples being used for training the model and the other 30% for validation. Specifically, $M_{13}$ and $G_{13}$ denotes the differences in the MODIS and GF-1 images on the date of $T_1$ and $T_3$. The K-fold cross-validation method is utilized to assess the performance of the ERTFM.

Secondly, ERT Fusion Method Building. A large number of regression trees were generated by randomly splitting nodes, and the outputs of trees were aggregated to predict the final value. The coarse difference images were used as predictors, and the fine difference images were used as target variables. We also used the grid search function in the Sklearn library to select the optimal parameters. The number (N_Estimators) and the maximum depth (max_Depth) of regression trees were chosen to minimize bias.

Thirdly, Prediction and Reconstruction. The trained ERTFM was used to predict the fine difference images $G_{12}$ and $G_{23}$ from the coarse images $M_{12}$ and $M_{23}$. Then, an adaptive local weighting approach was adopted to reconstruct the fine images $G_2$. The weight functions can be written as:

$$G_2 = W_1 * (G_1 + G_{12}) + W_3 * (G_3 + G_{23}) \tag{1}$$

where $W_1$ and $W_3$ are the weighting functions for the image of $G_2$ from $G_1$ and $G_3$, respectively. The difference of images between the known date and the prediction date is employed to calculate the contribution of each prediction. The shorter the time span

between $T_1$ and $T_2$, and between $T_2$ and $T_3$, the greater the weight parameter of the images. In addition, a sigmoid function was used to compute the time weight parameters:

$$W_1 = 1/1 + e^{k*i} \tag{2}$$

$$i = |M_{23} - M_{12}| \tag{3}$$

$$W_3 = 1 - W_1 \tag{4}$$

Finally, there was a series of GF-like image predictions based on the available MODIS images. NDVI was then calculated according to the predicted reflectance, which can be combined with meteorological observation data to estimate the LE product with high spatiotemporal resolution.

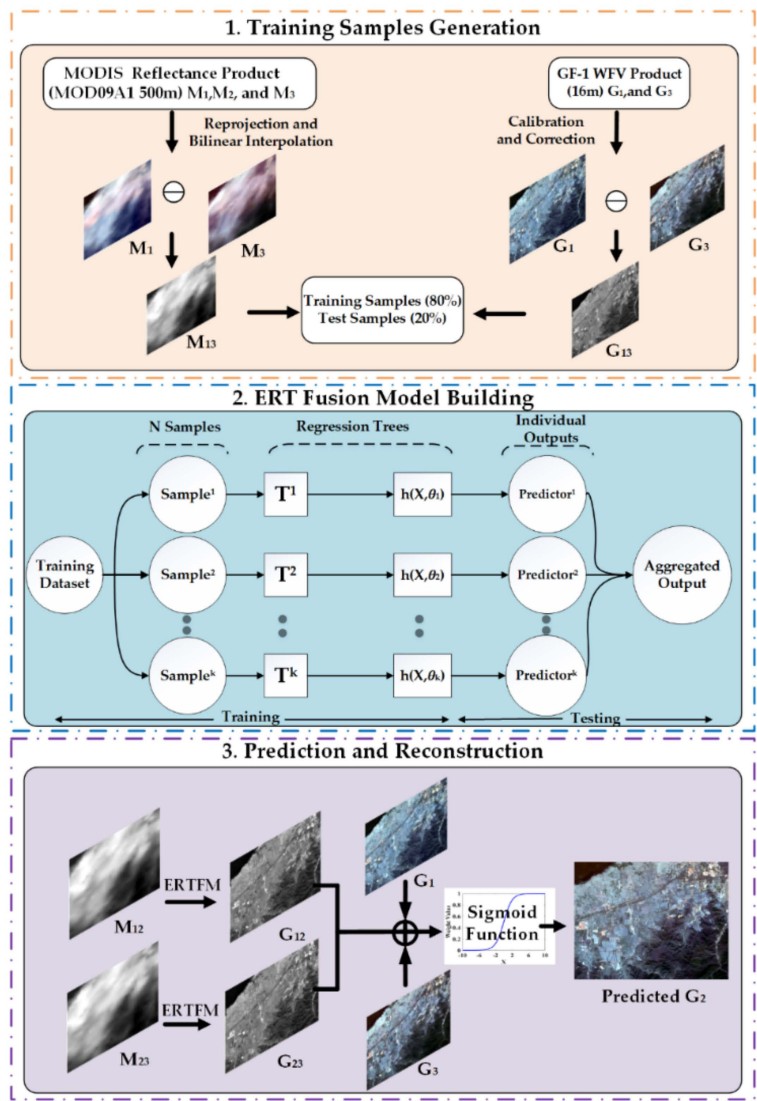

**Figure 1.** Flowchart of the proposed ERTFM.

*2.2. Comparison to Other Fusion Methods*

2.2.1. STARFM

The STARFM algorithm, the most widely used spatiotemporal fusion method, was first proposed by Gao et al. [19] to blend the MODIS and Landsat images. Given that the GF-1 and MODIS images have comparable bandwidth, STARFM can be expanded to yield GF-1 and MODIS fusion applications. It is assumed that the land cover types between the available date and predicted date do not change. Due to the complexity

and heterogeneity of land surface, the mixed pixels still exist in the coarse MODIS images. Therefore, STARFM introduced the neighborhood pixel information and utilized the weight function to calculate the reflectance of the center pixel.

### 2.2.2. ESTARFM

To improve the accuracy of STARFM over heterogeneous landscapes, the ESTARFM algorithm designed by Zhu et al. [20] introduced the conversion coefficient to calculate the changes between the fine-coarse pixels. Compared to the original model, the ESTARFM needs two periods of GF-1/MODIS images (at the same or a close date) and one additional MODIS image (at the predicted date) to compute the weighting parameter, which is especially suitable for a highly heterogeneous spatial area.

### 2.2.3. FSDAF

The Flexible Spatiotemporal data Fusion (FSDAF) method is a hybrid fusion method combining the unmixing theory and weighting function proposed by Zhu et al. [35]. The FSDAF can effectively predict the land cover change and needs just one pair of fine-coarse images, the first at the date of $t_0$ and the other coarse image at the date of $t_p$, to predict the target image. In our study, the FSDAF classified the GF-1 image first and calculated the temporal change of each class. Secondly, the residuals from temporal change were estimated. Then, a thin plate spline (TPS) method was adopted to downscale the coarse image, which was then combined with residuals to obtain the robust prediction. Finally, the weighting parameter was utilized to estimate the target pixels.

### 2.2.4. Fit-FC

Fit-FC method was firstly developed to blend the Sentinel-2 and Sentinel-3 images proposed by Wang and Atkinson [32], which consisted of regression fitting, spatial filtering and residual compensation. The most significant improvement of Fit-FC is an increase in the relationship between the available and predicted images. It also shows a robust performance in dealing with cases where substantial temporal changes occurred within a short time span. In the three steps, the linear fitting and spatial filtering are feasible approaches to eliminate the "patches" caused by the resolution differences. Finally, the residuals compensated for the predictions, and the spectral information of the fine image is preserved.

### 2.3. LE Computation

In this study, GF-1 and MODIS LE products were generated based on a Modified-Satellite Priestley–Taylor (MS–PT) algorithm proposed by Yao et al. [44]. It is developed by the classical Priestley–Taylor equation [45], which is designed for open water and saturated land:

$$LE = \alpha \frac{\Delta}{\Delta + \gamma}(R_n - G) \tag{5}$$

where $\alpha$ is the Priestly–Taylor coefficient (1.26), $\gamma$ is the psychrometric constant (0.066 kPa C$^{-1}$), $\Delta$ is the slope of saturation water vapor pressure versus temperature curve, $R_n$ is the net radiation, and $G$ is the soil heat flux. Based on the Priestley–Taylor type models [46], the MS–PT model utilized the apparent thermal inertia (ATI) to replace relative humidity and saturated vapor pressure to minimize the uncertainties of surface resistance, which has been verified to be effective in China [44,47]. The input parameters merely require the air temperature (Ta), the net radiation (Rn), diurnal air temperature range (DT) for ATI, and the vegetation index (NDVI). The terrestrial LE consists of the unsaturated soil evaporation ($LE_s$), the canopy transpiration ($LE_c$), the canopy interception evaporation ($LE_{ic}$), and the saturated wet soil surface evaporation ($LE_{ws}$). The MS–PT algorithm can be given by:

$$LE = LE_s + LE_c + LE_{ic} + LE_{ws}, \tag{6}$$

$$LE_s = (1 - f_{wet})f_{sm}\alpha\frac{\Delta}{\Delta + \gamma}(R_{ns} - G), \tag{7}$$

$$LE_c = (1 - f_{wet})f_v f_T \alpha\frac{\Delta}{\Delta + \gamma}R_{nv}, \tag{8}$$

$$LE_{ic} = f_{wet}\alpha\frac{\Delta}{\Delta + \gamma}R_{nv}, \tag{9}$$

$$LE_{ws} = f_{wet}\alpha\frac{\Delta}{\Delta + \gamma}(R_{ns} - G), \tag{10}$$

$$f_{sm} = ATI^k = \left(\frac{1}{DT}\right)^{DT/DT_{max}}, \tag{11}$$

$$f_{wet} = f_{sm}{}^4, \tag{12}$$

$$f_c = \frac{NDVI - NDVI_{\min}}{NDVI_{max} - NDVI_{\min}}, \tag{13}$$

where $f_{wet}$ is the relative surface wetness, $f_{sm}$ is soil moisture constraint, $f_T$ represents plant temperature constraint $(\exp(-(T_a - T_{opt})/T_{opt})^2)$, $DT_{max}$ describes the maximum diurnal air temperature range (40 °C), $T_{opt}$ is an optimum temperature (25 °C), $R_{ns}$ is the surface net radiation to the soil ($R_{ns} = R_n(1 - f_c)$), $G$ is soil heat flux ($\mu R_n(1 - f_c)$, $\mu = 0.18$), $R_{nv}$ represents the surface net radiation to the vegetation ($R_{nv} = R_n f_c$), $f_v$ is the vegetation cover fraction, and $NDVI_{min}$ and $NDVI_{max}$ are the minimum and maximum $NDVI$, respectively.

Compared with the MODIS LE product (MOD16), the MS–PT algorithm reduced the uncertainties by approximately 5 W/m² in the LE estimation and provided more reliable LE estimations at multiple biomes [14]. Therefore, the MS–PT was chosen to generate a spatiotemporal continuous LE product with 16 m daily resolution.

### 2.4. Assessment Metrics

The following statistical metrics were used to assess the accuracy of the ERTFM approach: the structural similarity (SSIM), the correlation coefficient (r), and average absolute deviation (AAD). SSIM denotes the similarity between the fusion image and true image, ranging from zero to one. The higher the SSIM value is, the better performance is. R denotes the correlations between observations and predictions. AAD measures the absolute deviation.

### 2.5. Experimental Data and Preprocessing
### 2.5.1. Study Area

The study area (14 km × 11 km) was selected near Guanting Reservoir, which is located in Huailai County, Hebei Province of Northern China (Figure 2). It is covered with a variety of heterogeneous landscapes, including cropland, forest, grassland, shrubland, wetland, water, impervious surface, and bare land, thus allowing for the assessment of the ETRFM model under different conditions. The study area has a typical continental climate with an average annual temperature of 10 °C and an average annual precipitation of 500 mm. This area contains two flux towers surrounding the fields with irrigated corn: Site 1 (115.788°E, 40.3491°N; hereafter EC 10 m site) and Site 2 (115.7923°E, 40.3574°N; hereafter EC 40 m site). The EC system, equipped with both a gas analyzer and a sonic anemometer, provided flux measurements ranging from 2014 to 2017, with a temporal resolution of 30 min. In this study, all flux measurements have been time-averaged to a daily value and any half-hourly measurements with low quality (QC > 2) were excluded. Any day with gaps of more than 25% of the entire time were indicated as missing. After that, the energy balance closure across two sites was 79.2% on average. Because of the energy non-closure limitation, the widely used method proposed by Twine et al. [48] was used for energy balance closure [49,50].

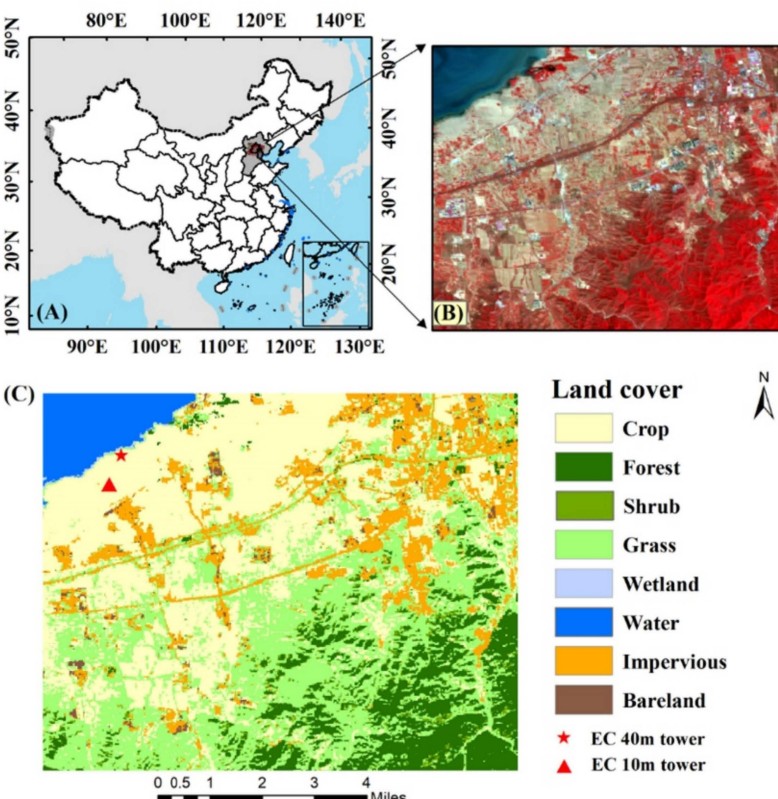

**Figure 2.** Maps showing: (**A**) The location of the study area; (**B**) GF-1 image collected on 2017/133; (**C**) Land cover types and the location of EC towers.

### 2.5.2. Remotely-Sensed Data

The Chinese GF-1, as the first satellite of the China High Resolution Earth Observation System, has the advantages of high spatial resolution, multispectrality, and large width [51]. The GF-1 spectrum ranges from 450 to 890 nm, which contains four spectral channels: bands 1 (blue), 2 (green), 3 (red) and 4 (near-infrared). Due to the unfavorable atmospheric conditions with cloud and aerosol contamination, images with a spatial resolution of 16 m and a temporal resolution of 4 day were frequently absent. In this paper, only nineteen cloud-free GF-1 WFV images were obtained during the period of 2014–2017. The systematic atmospheric, radiation and geometric corrections were processed. In addition, the Landsat 8 OLI product was used to correct the positioning accuracy of GF-1 images.

The MODIS spectral reflectance product (MOD09GA) contains bands 1–7 surface reflectance with a 500 m daily resolution, and was provided by the Land Processes Distributed Active Archive Center (LP DAAC). It has corrected the effects of atmospheric gases and aerosols and gridded these factors into the sinusoidal projection. The four spectral bands, including bands 1 (red), 2 (near-infrared), 3 (blue) and 4 (green), were selected and reordered 3 (blue), 4 (green), 1 (red), and 2 (near-infrared) to match the GF-1 bands. The MODIS Reprojection Tool (MRT) was used to convert the file format and reprojection [52]. Then, the MODIS product was resampled to the 16 m resolution using the bilinear interpolation method to meet the model input requirement. In addition, the outliers caused by cloud shadow contamination were removed. Since we have highlighted acquiring spatially and temporally continuous LE products, the cloud-free image close to the acquisition dates provided complementary information to fill in the gaps due to missing pixels.

### 2.5.3. Auxiliary Data

The meteorological dataset was obtained from the automatic weather station (AWS), which provides 10 min measurements of air temperature (Ta), wind speed (Ws), relative humidity (RH), soil heat flux (Gs), and shortwave solar radiation (Rs). All meteorological

variables were aggregated into daily or eight-day mean values. To ensure that records were only retained for dates with high-quality forcing data, all meteorological measurements along with flux measurements from EC systems corresponding to rain events and snow conditions were treated as missing. On the other hand, the shortwave broadband albedo was acquired from the Global LAnd Surface Satellite (GLASS) albedo product (5 km, 8 day) [53], which has satisfactory spatial-temporal coverage and reasonable consistency with ground measurements. The digital elevation model (DEM) data was acquired from the 90 m Shuttle Radar Topography Mission (SRTM) images, which were employed to calculate surface Net Radiation (Rn) in this study.

## 3. Results

### 3.1. Evaluation of the ERTFM

Four GF-like images were produced by the ERTFM based on the available GF-1 and MODIS reflectance products. Four actual observed GF-1 images captured on the dates of 2015/269, 2016/112, 2016/121, and 2017/129 served as reference data for the independent assessment. Figure 3 provides the visual comparison between the actual GF-1 images and predictions by the ERTFM on validation dates. Compared with the original images, the images fused by the ERTFM can maintain spatial details and closely resemble the actual image. Although the land surface is highly heterogenous, it is apparent that the ERTFM algorithm was able to capture the villages, towns, roads, and sparse patches of vegetation, and achieved a satisfactory reconstruction.

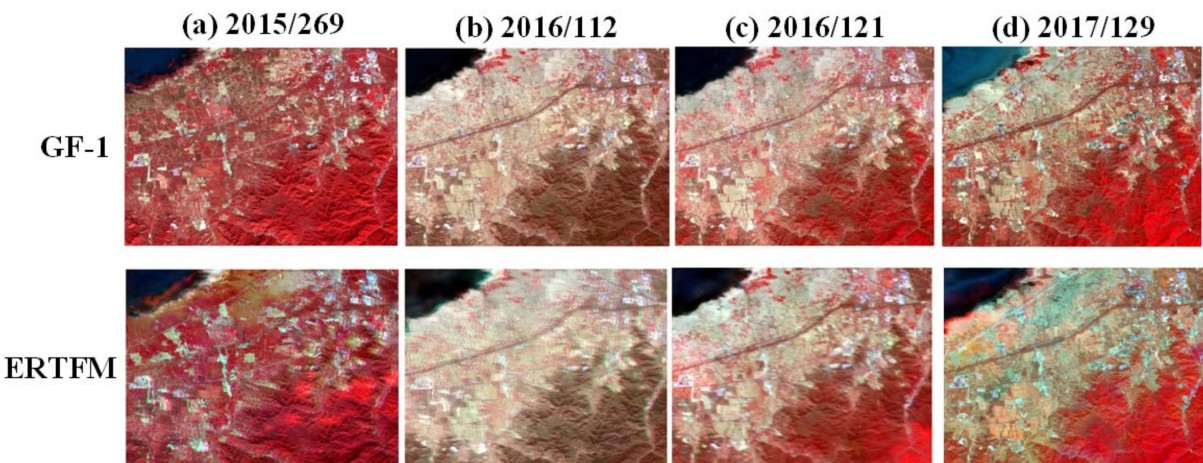

**Figure 3.** The actual GF-1 images (RGB composite) and the predictions by the ERTFM.

Table 2 presents the quantitative assessment of four predicted bands and shows that the ERTFM-based images are similar to the actual observed GF-1 images, with higher values of SSIM and r, and lower values of RMSE and AAD. The scatter plots of comparison between the predicted and observed bands (as shown in Figure 4) also demonstrated that the ERTFM performed well. The reflectance after fusion shows a high correlation with the observed GF-1 and most of the values show a one-to-one relationship. For all prediction bands, SSIM ranged from 0.86 to 0.96, r was between 0.7 and 0.83, and the RMSE and AAD were in acceptable ranges from 0.012 to 0.032. The blue and green bands were more similar to the actual GF-1 band compared with the red and near-infrared (NIR) bands in terms of SSIM, while the red and near-infrared bands provided higher correlations than other bands. The overall validation indicates that the ERTFM is accurate enough to produce satisfactory estimations.

To further investigate the performance of the ERTFM, we compared the predicted NDVI, calculated by the red and NIR bands, and the actual GF-1 NDVI (Figure 5). The results show that the synthetic NDVI also presented close agreement with the observed NDVI, r ranged from 0.76 to 0.86, RMSE was around 0.11, and AAD ranged from 0.08

to 0.1. It is apparent that in the growing season (2016/121, 2017/133 and 2016/269), the predictions by the ERTFM achieved higher accuracy. Although the fusion of raw reflectance bands would appear to have introduced errors, such as the anomalous values in NIR reflectance, the advanced filters by calculating NDVI could eliminate the outliers and improve the accuracy. The NDVI is regarded as one commonly used parameter to monitor water and energy cycling. Therefore, it is feasible to apply the predicted reflectance data to calculate terrestrial parameters, such as the vegetation index, and obtain satisfactory and reasonable predictions.

**Table 2.** The accuracy assessment of four fusion bands by ERTFM between the actual GF-1 images and predicted images.

| Date | Band | SSIM | RMSE | AAD | r |
|------|------|------|------|-----|---|
| 2015/269 | Band1 | 0.938 | 0.016 | 0.012 | 0.74 |
| | Band2 | 0.888 | 0.024 | 0.021 | 0.69 |
| | Band3 | 0.868 | 0.028 | 0.021 | 0.72 |
| | Band4 | 0.91 | 0.028 | 0.021 | 0.7 |
| 2016/112 | Band1 | 0.96 | 0.016 | 0.012 | 0.71 |
| | Band2 | 0.937 | 0.018 | 0.021 | 0.74 |
| | Band3 | 0.896 | 0.027 | 0.021 | 0.78 |
| | Band4 | 0.89 | 0.031 | 0.021 | 0.83 |
| 2016/121 | Band1 | 0.968 | 0.015 | 0.012 | 0.71 |
| | Band2 | 0.949 | 0.017 | 0.021 | 0.74 |
| | Band3 | 0.9 | 0.027 | 0.021 | 0.78 |
| | Band4 | 0.904 | 0.032 | 0.021 | 0.83 |
| 2017/129 | Band1 | 0.922 | 0.02 | 0.017 | 0.81 |
| | Band2 | 0.906 | 0.019 | 0.014 | 0.76 |
| | Band3 | 0.863 | 0.025 | 0.019 | 0.82 |
| | Band4 | 0.866 | 0.032 | 0.025 | 0.8 |

*3.2. Comparison with Other Fusion Methods*

A visual comparison of the predictions via the five methods and the actual image showed that each of these methods correctly captured the fine spatial information and general temporal change (Figure 6). It is apparent that the predictions are generally similar to the original GF-1 image and present the GF-like images. Compared with the other four methods, the image produced by the ERTFM more closely resembled the observed GF-1 image, and the boundaries between roads, villages, and patches of vegetation are clearly distinguished. In contrast, the Fit-FC predictions appear blurry, with less clear boundaries, especially in farmland areas. Due to the mixed medium resolution pixels in the MODIS image, the great heterogeneity and rapid temporal changes generally resulted in an unsatisfactory performance. The images fused by FSDAF and STARFM produced outliers in some pixels, with the result that the images were relatively fuzzy. Therefore, the ERTFM was able to predict images that more closely resembled the actual GF-1 than the other four methods.

Scatter plots (Figure 7) and quantitative indicators showed that all the fusion methods can achieve satisfactory accuracy with the higher SSIM and r, and lower RMSE and AAD. Among the five fusion methods, the ERTFM results yielded the most accurate predictions, and the points were closer to the 1:1 line. The proposed ERTFM method had the highest SSIM (0.91) and r (0.79) values, and the lowest RMSE (0.023) and AAD (0.017) values, followed by ESTARFM and STARFM. For the specific band, the performance of the ERTFM

was also superior to the other four fusion methods. The correlation coefficient of the predicted red band and the NIR band can reach more than 0.81, while the accuracy levels of the other four fusion methods were much lower on the validation date. ESTARFM and STARFM have a comparable capacity to produce accurate results. Nevertheless, they made erroneous estimations for some pixels. In particular, the performance of the FSDAF and the Fit-FC leaves much to be desired and the obvious outliers are dispersed around the 1:1 line.

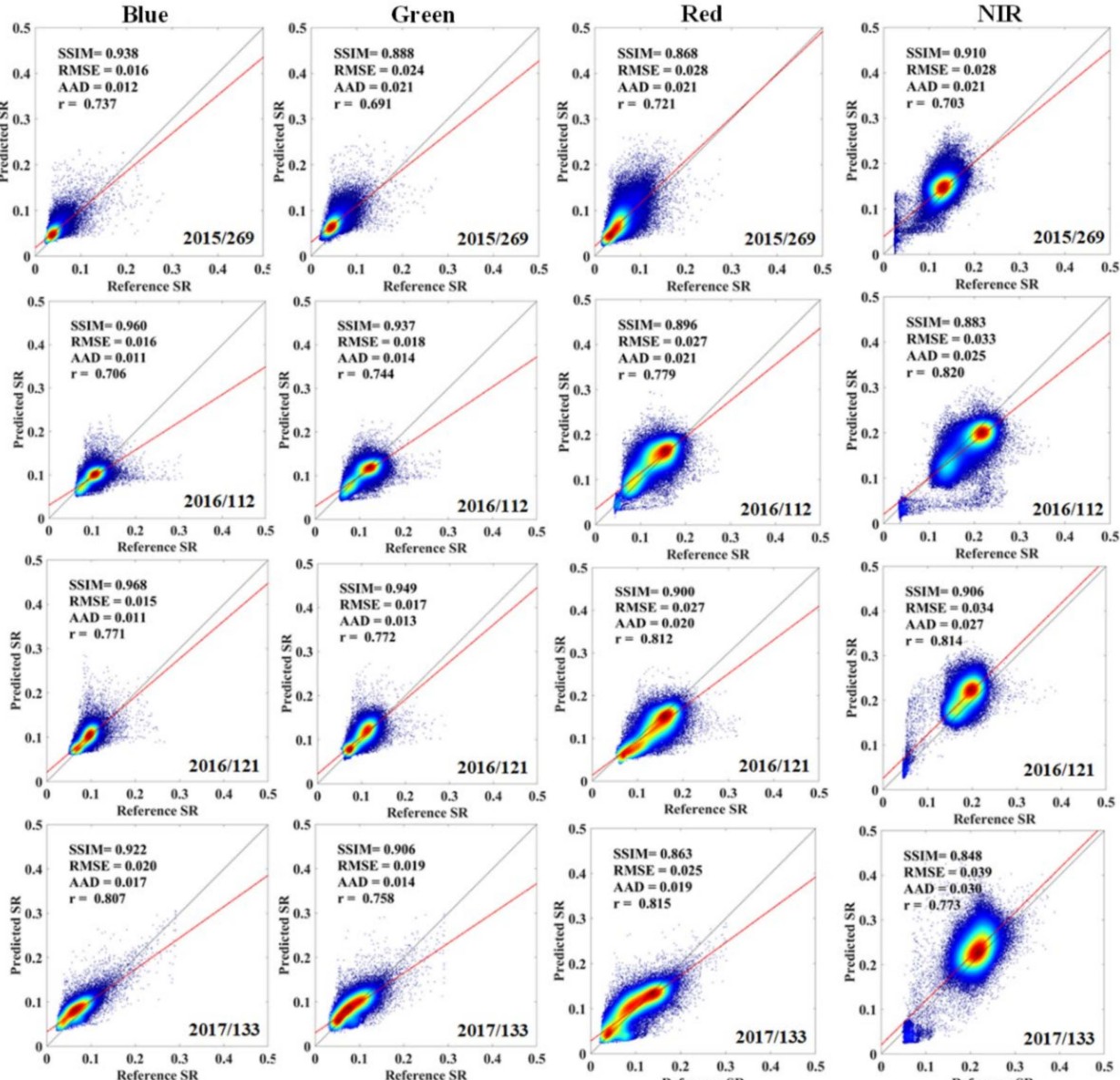

**Figure 4.** Scatter plots of predicted reflectance by the ERTFM and the actual GF-1 bands. The red line is the fitting line, and the black line is the 1:1 line.

According to the statistical parameters (Figure 8), we can see that with respect to SSIM (ranging from 0.889 to 0.93), r (ranging from 0.713 to 0.8), AAD (ranging from 0.017 to 0.019), or RMSE (ranging from 0.023 to 0.024), the ERTFM is superior to the other four fusion methods as far as the whole images are concerned. The accuracy of the ESTARFM is lower than the ERTFM, but higher than the others, with an SSIM value of 0.9, an r value of 0.75, an AAD value of 0.21 and an RMSE value of 0.024. The STARFM yields relatively lower SSIM (0.88) and r (0.712) values, and higher RMSE (0.026) and AAD (0.021) values. The Fit-FC yields the lowest validation accuracy compared with the other four methods (r varying from 0.543 to 0.699 and RMSE varying from 0.30 to 0.38).

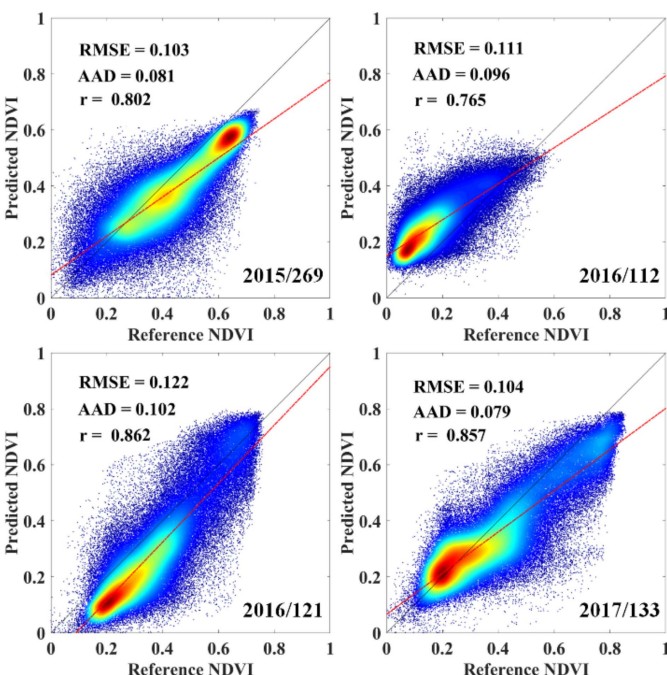

**Figure 5.** Scatter plots of predicted NDVI compared with GF-1 reference NDVI. The red line is the fitting line, and the black line is the 1:1 line. Negative pixels of the modelled NDVI corresponding to a body of water were not included in the statistical analysis.

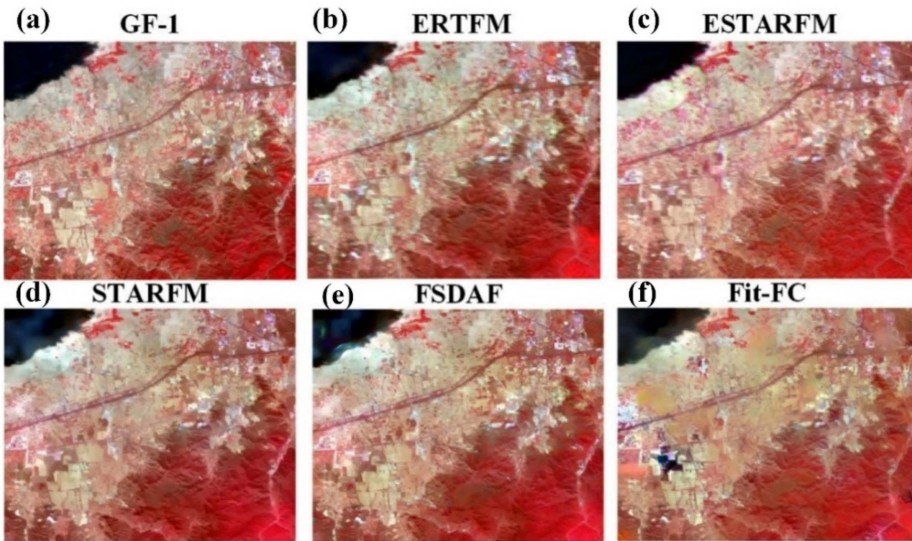

**Figure 6.** False color composites (RGB) of reflectance observed on the date of 2016/121 by: (**a**) GF-1; (**b**) ERTFM; (**c**) ESTARFM; (**d**) STARFM; (**e**) FSDAF; (**f**) Fit-FC.

### 3.3. The Application of the ERTFM on LE Estimation

To evaluate the performance of the ERTFM for estimating LE at high spatiotemporal resolution (daily and 16 m), we firstly selected six cloud-free GF-1 images and combined them with MODIS images at the corresponding date. Then, we applied the ERTFM model to produce temporally continuous reflectance data at high spatial resolution, which can be used to derive vegetation indices, such as NDVI. Based on the improved NDVI generated by the ERTFM and the original NDVI from MODIS, the spatial distributions of LE estimates during the growing season (May to September) in 2017 were presented in Figure 9 (ERTFM-applied) and Figure 10 (original MODIS), respectively.

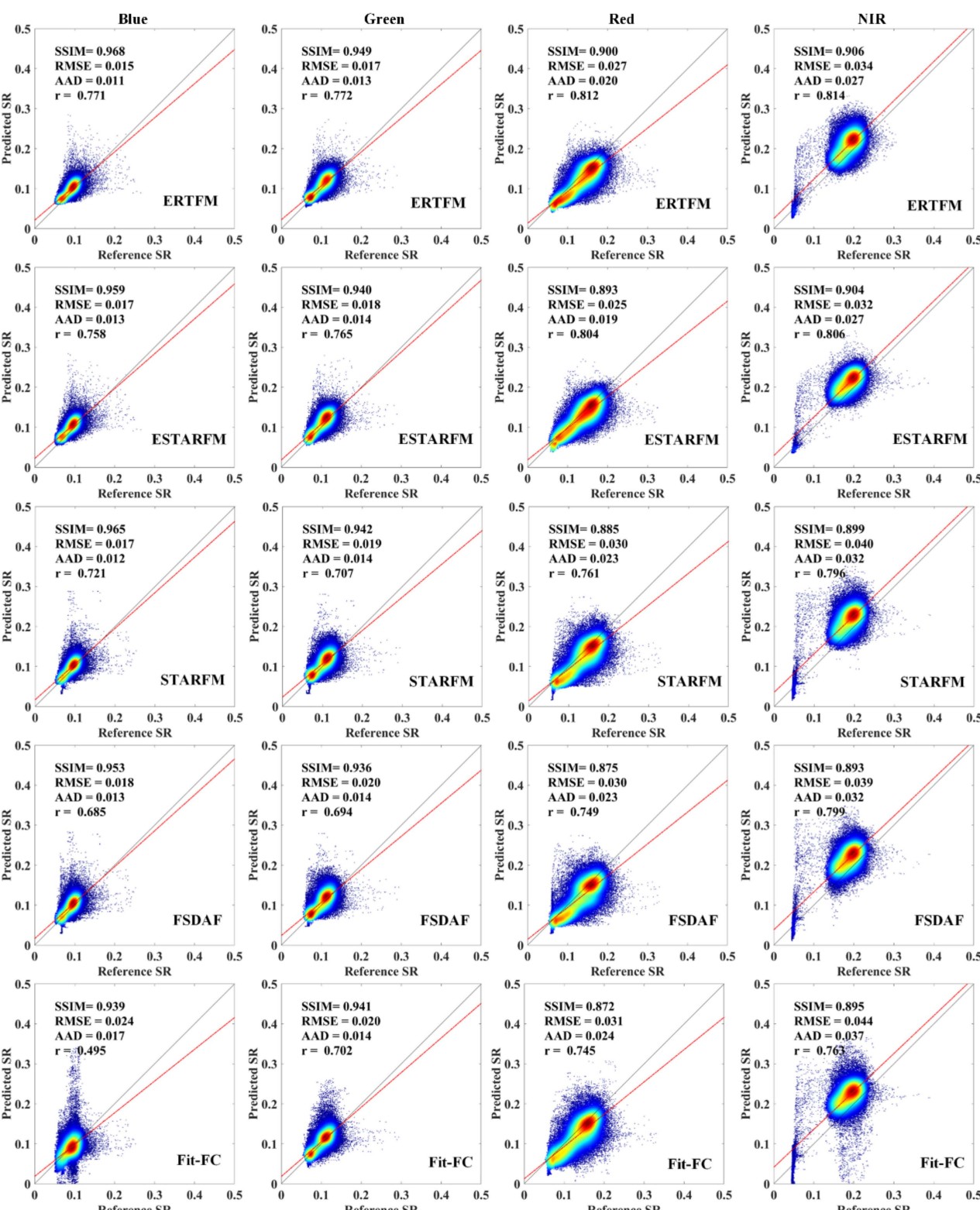

**Figure 7.** Scatter plots of observed GF-1 reflectance and predicted reflectance by ERTFM, ESTARFM, STARFM, FSDAF, and Fit-FC. The red line is the fitting line and the black line is the 1:1 line.

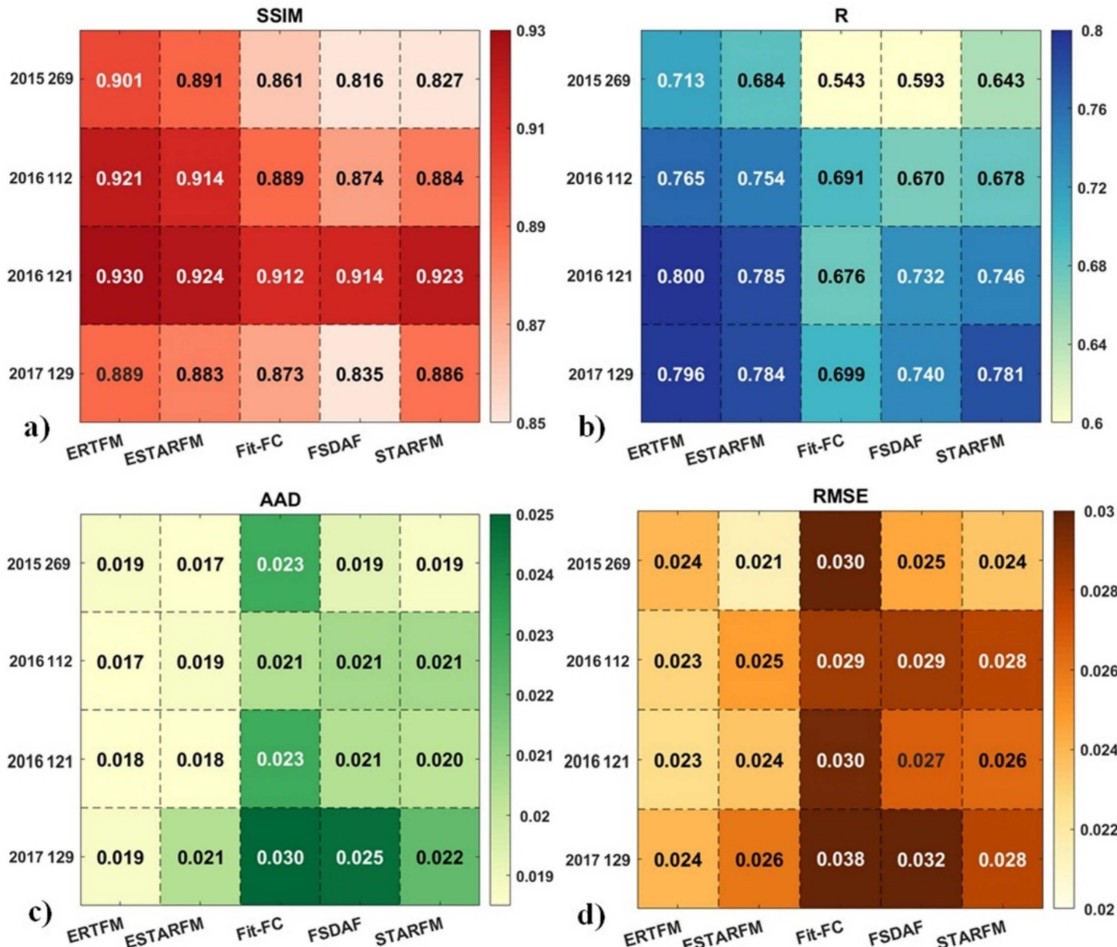

**Figure 8.** Comparison of the ERTFM with the other four spatiotemporal fusion models, STARFM, ESTARFM, FSDAF, Fit-FC. (**a**) SSIM. (**b**) r. (**c**) AAD. (**d**) RMSE.

Compared with Figure 10, LE estimates using the ERTFM-based NDVI had similar temporal variation characteristics but provided more details of spatial variances. Particularly for the period from June to August, the regional distribution of LE estimates differed significantly in the study area. In the beginning of May, LE estimates of different land covers were relatively lower, contributing to less spatial variances in both Figures 9 and 10. After the middle of May, LE estimates located in forested areas and surrounding villages significantly increased, due to rising temperature and solar radiation. By contrast, the increasing trend of LE estimates in croplands started in June, and gradually reached its peak value in July. That is because crop transpiration played an important role in the rising of LE temporal variations, and the agricultural managements, such as irrigation, can be regarded as an acceleration factor in this irrigated corn area. However, the simulated LE decreased on the 201st day of 2017 but increased again on the 209th day of 2017. This suggests that the 201st day of 2017 experienced a rain event, causing decreased solar radiation and further lowered LE. As autumn was coming, LE estimates in croplands show a decreasing trend from August to September, indicating that the corn gradually came to maturity with less transpiration. In general, LE estimates using the ERTFM-based NDVI provided more detailed spatial characteristics and showed clearer variations in terrestrial drought conditions—a feature of the method that would be of use in regional precision water management and agricultural management.

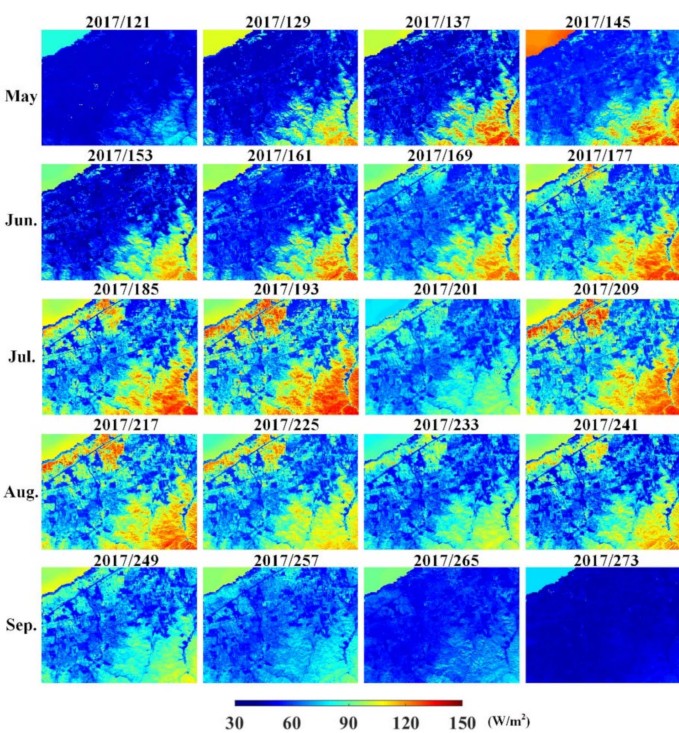

**Figure 9.** Spatial time series (every 8 days) of LE product based on the ERTFM in the growing season (from May to September 2017).

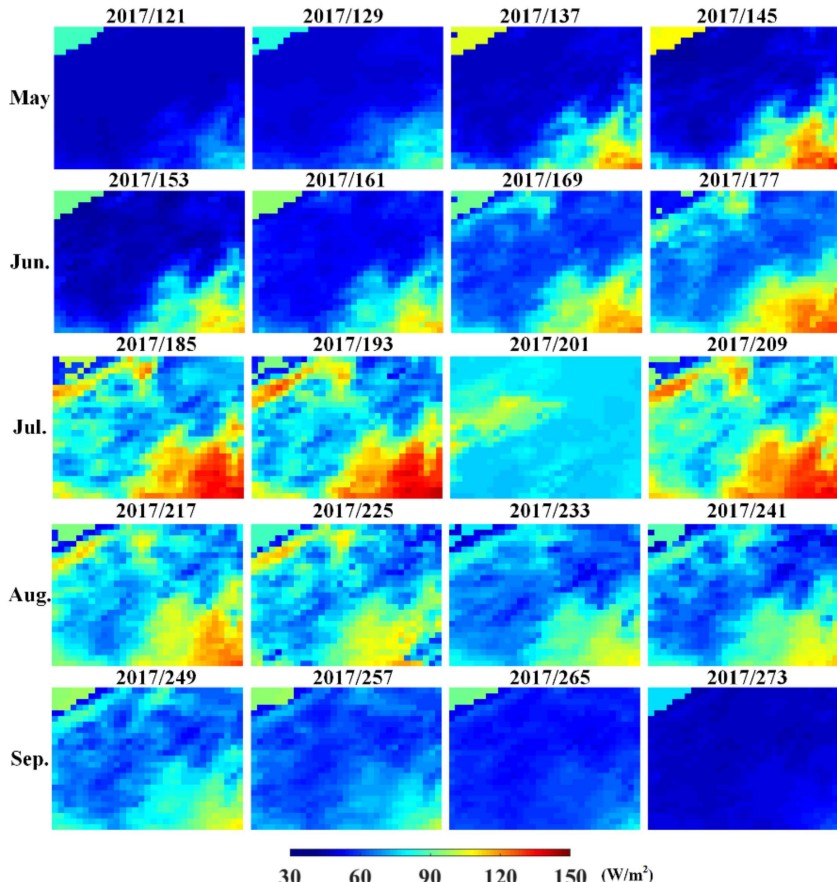

**Figure 10.** Spatial time series (every 8 days) of MODIS LE product before fusion in the growing season (from May to September 2017).

A field scale LE, taken over 8 days with 16 m resolution, was produced successfully based on the proposed ERTFM method. To further evaluate the performance after fusion, we compared the two satellite products with the ERTFM-based LE products and EC observations in Figure 11. The blue circle and orange circle represent the MODIS and GF-1-derived LE products, respectively. After fusion, LE estimation showed improved accuracy compared with the coarse MODIS product with an $R^2$ of 0.8 and an RMSE of 19.2 W/m$^2$. Before fusion, the MODIS and GF-1 LE products were slightly overestimated or underestimated. After fusion, the estimated LE and EC measurements showed closer agreement, and most of the pixels followed the one-to-one relationship. The RMSE of the predicted LE decreased from 23.0 to 19.2 W/m$^2$, which implies that the accuracy of the estimated LE has been improved slightly. Because the estimated LE was reconstructed by the fusion model based on the MODIS and GF-1 satellite images, there was limited improvement in accuracy. The simulated reflectance and NDVI were determined by MODIS and GF-1 data, therefore the accuracy of the estimated LE also depended on these two satellite products.

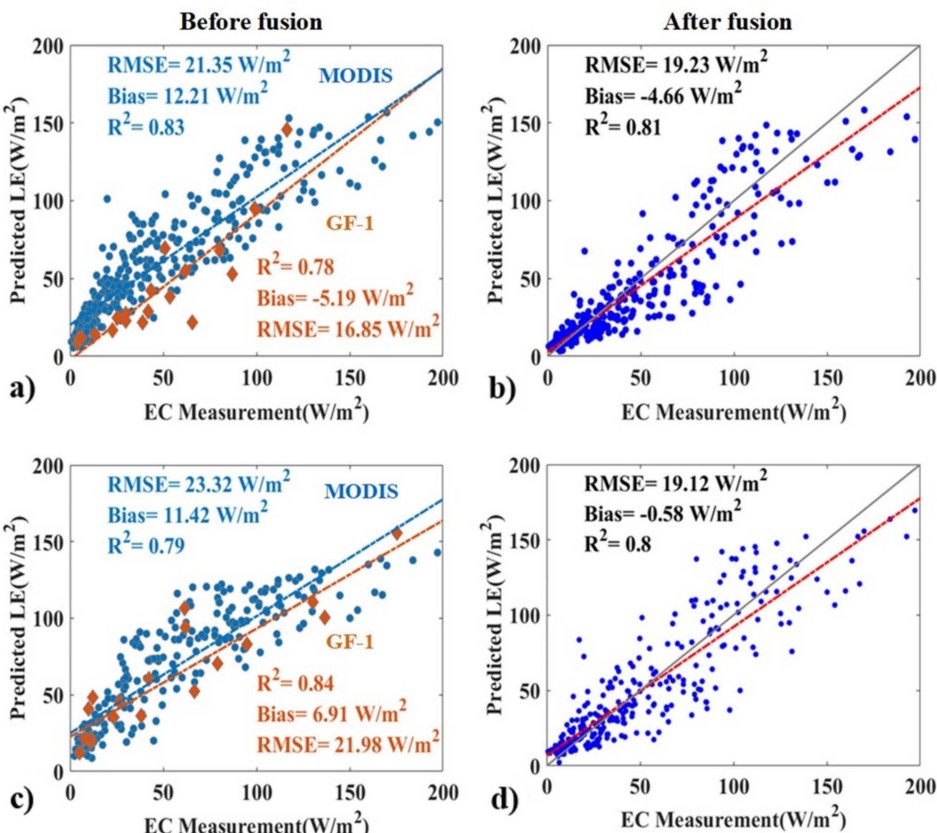

**Figure 11.** Scatter plots of MODIS LE, GF-1 LE, and integrated LE products validated by the EC measurements: (**a**,**b**) for EC 10 m site; (**c**,**d**) for EC 40 m site. The red line is the fitting line and the black line is the 1:1 line.

Figure 12 plots the time series comparison of MODIS LE, GF-1 LE and the ERTFM-based LE and EC ground measurements. The results indicated that the temporal variation trend of satellite LE and simulated LE products showed favorable agreement with in situ measurements. The interannual variation of LE follows one peak pattern and reaches its peak in summer from June to August, which corresponds to the continental monsoon climate. In the early and late stages of the growing season, the MODIS product generally overestimated LE with the bias of approximately 15 W/m$^2$, while the GF-1 based LE product showed closer agreement with EC measurements. The temporal trends of LE verify that there exists a significant complementarity in the temporal and spatial characteristics of the MODIS and GF-1 products. More importantly, the ERTFM-based LE was able to

combine the advantages of these two products and successfully captured the fine spatial and temporal details. The estimated LE also showed a good correspondence with the precipitation events. It is not surprising to find that the ERTFM-based LE showed the notable trough in the growth period (DOY 200), this having been caused by the decreased shortwave radiation in the cloudy and rainy conditions. Overall, the estimated LE based on the ERTFM was consistent with EC measurements and captured the rapid temporal changes that took place over the heterogeneous land surface.

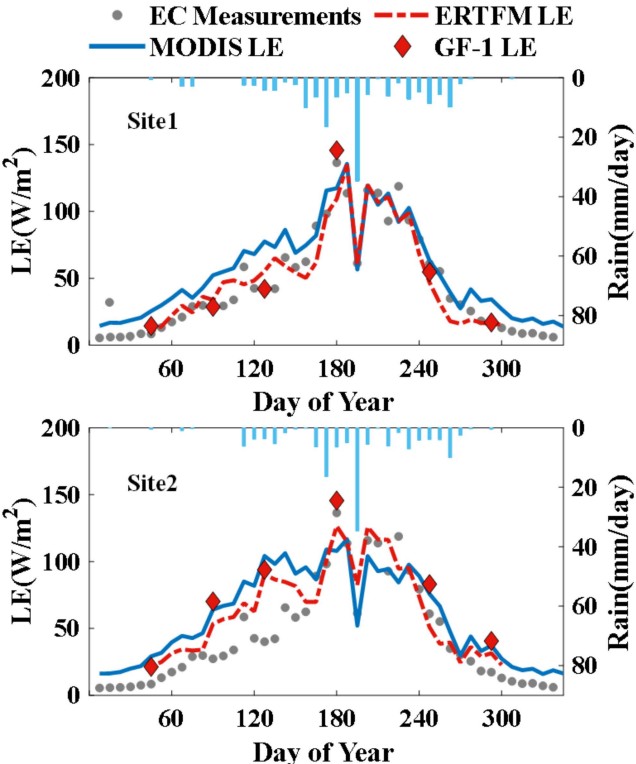

**Figure 12.** Time series of the MODIS LE (blue line), the GF-1 LE (red diamond), and the ERTFM-based LE (red line) products validated by the EC measurements (gray circles) on an 8-day average scale. Blue bars indicate rainfall events.

## 4. Discussion

### 4.1. Performance of the ERTFM

Our results demonstrated that the proposed ERTFM algorithm is able to merge GF-1 and MODIS reflectance imageries with higher values of SSIM and r, and lower values of bias and AAD. The feature mapping function constructed by the ERT machine-learning modeling between the coarse and fine resolution imagery is effective in predicting land surface reflectance with high spatiotemporal resolution. Among different spectral bands, the red and NIR bands showed relatively higher correlations than the other bands. This is likely due to a greater similarity in the scope of spectral band coverage between the GF-1 and MODIS sensors [51,54]. However, it is noted that some scatter points are dispersed and the overestimation or underestimation can be transferred to final predictions. The outliers and uncertainties can be explained in terms of the mixed MODIS coarse pixels leading to the smooth GF-1 extreme predicted values [15,55]. Given the complexity and variability of the land surface, the prediction errors would be amplified, thereby affecting the reflectance estimations [18]. Nevertheless, the spatiotemporal fusion method based on the ERT learning technique has shown that it is possible to build a nonlinear correlation using different images at the predicted and target times, without having recourse to the sophistication of sparse representation. Compared with other current fusion methods, the ERTFM presents the stronger generalization ability. Overall, the ERTFM algorithm

proved to be a feasible technique for improving the spatial and temporal resolution of multisource data, and was able to achieve a level of accuracy comparable to the other methods, even though the pixel-wise difference between the GF-1 and MODIS images is nearly thirty times.

The limitations of the ERTFM model include two major factors: (1) the accuracy of high precision images; (2) the reliability of training samples. Theoretically, spatiotemporal fusion requires precise image alignment to provide the exact same views [52]. Although the GF-1 and MODIS datasets have similar spectral configurations and band width, obvious differences still exist in different satellite sensors, such as the geometry registration and atmospheric conditions [56]. For example, the GF-1 satellite has the broader swath with larger viewing angles than MODIS, resulting in the different solar viewing angles and elevations. Therefore, it is necessary to preprocess the georeferencing and atmospheric corrections to ensure the accuracy of fusion methods. In our study, we utilized the same period of Landsat imagery to correct the distorted GF-1 images, but the manual adjustment is not enough to reach the fine-pixel bench level. On the other hand, the ERTFM model is employed to merge satellite datasets and achieve accurate predictions and reconstructions and thus the accuracy of the machine learning method is highly dependent on the quality of the training data samples. Previous studies found that the larger the amount of training samples, the higher the accuracy estimations [57]. However, a large number of training datasets would lead to low computing efficiency, particularly for a large-scale area. Therefore, the difficulty of striking a balance between the robustness of training samples and computing efficiency is the key factor limiting the application of the ERTFM model.

*4.2. Comparison with Other Fusion Models*

Several popular fusion algorithms have been employed for comparison using GF-1 and MODIS reflectance products in our study. According to our results, the overall fusion methods can successfully preserve the pixel-level spatial details and provide nearly unbiased predictions for each reflectance band. Among the five methods, the Fit-FC and FSDAF performed worse than the other methods due to an unsatisfactory blurring effect for great temporal variations. Some differences should be taken into account based on the actual experimental datasets, such as the input imagery, radiometric bias, and the principle of the algorithm.

Firstly, the large uncertainties are mainly attributed to the defectiveness of the prior fine-coarse image pair, which meant that less supplementary information was available. Previous studies also supported this finding, and suggested that the number of fine resolution images that are inputted has a significant influence on the performance of fusion methods [58]. Significantly, the fusion images utilized in our study are GF-1 and MODIS products, which are not identical to the products used in the other studies. It should be noted that the Fit-FC method was only recently developed for sentinel-2 and sentinel-3 fusion applications, and has more strict requirements [32]. Secondly, the performance of the fusion methods also depends on radiometric deviation. The Fit-FC and FSDAF are highly sensitive to radiometric inconsistences, but the biases in the different sensors are hard to eliminate completely [59]. Moreover, for the formulation of the Fit-FC algorithm, it first developed the relationship between the GF-1 and MODIS images using a linear regression, and further applied it to the target images. This strategy is not suitable for highly heterogeneous regions, which would explain why the Fit-FC algorithm produced the blocky effects that it did, and which resulted in the fuzzy boundaries. In contrast, the proposed ERTFM is easy to conduct and achieves a relatively higher level of accuracy. The extraordinary advantage of the ERTFM is the use of machine learning to replace the simple linear regression and provide a robust simulation.

The performance of the STARFM and ESTARFM methods is not as accurate as the proposed ERTFM, while the latter's ability for preserving the GF-like image structures outperforms the Fit-FC and FSDAF methods. STARFM, as the most popular used fusion algorithm, utilizes information from neighboring pixels with similar spectra to predict the

missing data [60]. However, this algorithm is based on the assumption that one MODIS pixel contains one land cover type, which would be problematic for heterogeneous pixels. According to our results, STARFM produced much less accurate estimations, especially in conditions of high heterogeneity and temporal variation. A similar comparison was performed by Gevaert and García-Haro [15], who suggested that STARFM is more sensitive to temporal change than spatial variance. Regarding this issue, ESTARFM introduced a conversion coefficient to enhance the accuracy of prediction for heterogeneous landscapes. This modification led to a better performance (ΔR of +3.8%) of ESTARFM than STARFM in our study, which is consistent with the findings of Zhu et al. [20]. It should be noted that although ESTARFM is generally an "enhanced" version of STARFM, it does not always produce lower errors than STARFM, particularly for spectral bands where temporal variance is dominated [61]. Land cover with different domain characteristics and spatiotemporal variances was found to be strongly associated with the performance of STARFM-like methods [59].

*4.3. The Application of ERTFM*

According to our study, the ERTFM can be used to produce accurate and high spatiotemporal resolution LE by fusion of the GF-1 and MODIS products. This method was also validated with respect to the efficient capture of the spatial and temporal variability in LE at GF-like resolution. This fine LE information, if generated operationally, could assist with irrigation managers to arrange irrigation quantities and timing. For instance, Ma et al. [4] used field-scale LE to monitor irrigation water efficiency and suggested that the reallocation of irrigation water is needed to reduce wasted water that cannot be used by crops. Furthermore, given that NDVI is an indicator of plant phenological stage, many studies have verified the utility of high spatiotemporal resolution NDVI images for crop growth monitoring and yield prediction [62,63]. Due to the direct relationship with plants' carbon assimilation, LE estimates at field scale were also proven to be associated with within-field variability in at-harvest yield maps [37]. Thus, the superiority of the ERTFM when it comes to providing detailed LE information that may not be easily acquired with other methodologies means that the model promises a more comprehensive understanding of how LE relates to surface water stress and plant growth.

The ERTFM is an application of the ERT method to predict terrestrial LE, which could also be employed for other land surface parameters. For example, Zhang et al. [64] used eight machine learning regression algorithms to estimate forest aboveground biomass and found the recently developed ERT and CatBoost methods achieved better performances, providing more stable results. Similar studies were conducted to predict streamflow [65] and air quality [66]. These studies demonstrated that the ERT method not only provided solid performance on different datasets, but also achieved a good compromise between predictive accuracy and computational requirements. As noted by Shang et al. [43], the superior fusion performance of the ERT method could be attributed to the removal of the need for the optimization of discretization thresholds. More applications of the ERT method to generate images with high frequency and high spatial resolution are encouraged to better understand its pros and cons.

**5. Conclusions**

The accurate estimation of terrestrial LE is fundamental to regional water stress monitoring and water resources management. Despite the rapid development of new satellite-derived LE products, it is still a challenge to observe LE with both high frequency and high spatial resolution. Alternatively, several satellite-based data fusion approaches have been developed to bridge the gap between the high and low spatial resolution data and improve the spatiotemporal continuity of products. In this study, we combined the machine learning-based ERTFM and vegetation-based MS–PT method to produce high spatiotemporal resolution LE data by merging the Chinese GF-1 and MODIS products.

The validation results illustrated that the fused reflectance data generated by the ERTFM showed a close agreement with the fine GF-1 data (SSIM = 0.91, r = 0.76, RMSE = 0.023, AAD = 0.02) and presented the distinct features of roads and agricultural areas. The NDVI calculated from the fused reflectance data also had a high correlation (r = 0.83) with GF-1 NDVI. Through a comparison with four fusion models including STARFM, ESTARFM, FSDAF, and Fit-FC, ERTFM was proven to explain more observed reflectance variances with a mean r value of 0.79, followed by the ESTARFM (r = 0.75) and STARFM (r = 0.71). Further analysis demonstrated that high spatiotemporal resolution LE estimates calculated from the fused reflectance data agreed well with ground LE measurements ($R^2$ = 0.81, RMSE = 19.18 W/m$^2$). Our findings indicated that the ERTFM is able to improve ET estimates by integrating daily information at moderate resolution from wide-swath sensors like MODIS, with periodic high spatial resolution images from GF-1. Spatiotemporally continuous LE estimates of a high quality are projected to support agricultural monitoring and irrigation management.

**Author Contributions:** Conceptualization, Y.Y., X.B. and L.Z.; methodology, K.S.; software, J.Y.; validation, X.B., Y.L. and L.Z.; formal analysis, X.B.; investigation, X.G.; resources, R.Y.; data curation, Z.X.; writing—original draft preparation, X.B. and L.Z.; writing—review and editing, Y.Y. and L.Z.; visualization, X.B.; supervision, Y.Y.; project administration, Y.Y.; funding acquisition, Y.Y. All authors have read and agreed to the published version of the manuscript.

**Funding:** This research was funded by the National Key Research and Development Program of China (Grant No. 2016YFA0600103 and No. 2016YFA0600102), the National Natural Science Foundation of China (Grant No. 41671331). This work was also supported by the China Scholarship Council under Grant 201806040217 and the Natural Resources Department of ITC, University of Twente, Netherlands.

**Institutional Review Board Statement:** Not applicable.

**Informed Consent Statement:** Not applicable.

**Data Availability Statement:** Not applicable.

**Acknowledgments:** The authors thank Ziwei Xu and Zhongli Zhu from Beijing Normal University, China, for providing ground-measured data. We would like to thank Bing Wang and Ning Hou from Beijing Normal University, China, for help with the programming. We would also like to acknowledge the data support from "National Earth System Science Data Center, National Science & Technology Infrastructure of China (http://www.geodata.cn, accessed on 15 October 2020)". The Chinese GF-1 WFV data was obtained from the Ministry of Ecology and Environmental Center for Satellite Application on Ecology and Environment (http://www.secmep.cn/, accessed on 10 September 2020). The MODIS surface reflectance product was downloaded online (http://glovis.usgs.gov/, accessed on 9 September 2020). The land cover data was downloaded online (http://data.ess.tsinghua.edu.cn/, accessed on 9 Septembar 2020). The digital elevation model (DEM) data was obtained from 90 m Shuttle Radar Topography Mission (SRTM) images (version 004) (http://srtm.csi.cgiar.org/, accessed on 2 October 2020).

**Conflicts of Interest:** The authors declare no conflict of interest.

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
