# Peer review of "ERTFM: An Effective Model to Fuse Chinese GF-1 and MODIS Reflectance Data for Terrestrial Latent Heat Flux Estimation"

_remotesensing, doi:10.3390/rs13183703_

Round 1
Reviewer 1 Report
This study attempts to propose an approach to fuse multiple satellite remote sensing observations by leveraging their strengths in high resolution and revisits to produce high spatial and temporal resolutions to estimate Latent heat fluxes over a chosen region. Compared with previous approaches that are essentially linear because of their dependence on simple regressions, this approach uses machine learning to replace the regression with the Extremely Randomized Trees (ERT) which potentially offers a non-linear approach. Evaluation of the results showed that the proposed approach, the ERTFM, showed the highest skill to capture latent heat flux dynamics for the chosen period over both space and time.
There are a few points that I think will make the manuscript and the full potential of the approach be better articulated to help potential users.
Comments
- Title: I think the word “Robust” is used too lightly here. Robustness is aimed at showing the quality would not change in spite of how and where it is applied. I don’t think the application domain is sufficient to render the method robust yet. I think this is overreaching.
- On the same matter, could you include a section in the discussion that captures the limitations of the proposed approach? I think that needs to be clear. It is very alright that any approach would have limitations. Any good Science would highlight its flaws allow for a more efficient use.
- Line 58: “However, due to the lack of detailed information description,…” Please clarify what is meant by “lack of detailed description”.
- Lines 142-143: “There are three key parameters in the procedures K, Nmin and M, which can be automatically adjusted according to the learning samples.” I was quite excited to see this section, however, being such an important statement, it played NO followup role in helping us understand its benefits throughout the rest of the manuscript. How do changes in these parameters impact the resulting quality of ERTFM? And how should users think of tuning them for better results.
- Lines 145-147: How is the problem of over-fitting dealt with? Please include a brief description.
- I hope I haven’t missed any information in the LE computation in Section 2.3. A couple of concerns come to mind. Firstly, alpha and gamma in the PT equation are not defined. Please do so. Secondly, the aim of the use of the formalism from this study is to produce high spatial resolution LE products. In this regard, I think alpha especially cannot be constant. It would change from place to place. I am not entirely sure of how alpha is estimated here. Please give more details to this and if a constant value were used, it would be fitting to provide reasons why a constant value was chosen.
- Section 2.5.1 presents the introduction of the study area and ground observations used. I believe it would be useful to discuss why this study area was chosen. What is the scientific benefit for the choice of this region? How do the characteristics of this region help to advance the objectives of this study? And what are the possible differences in results one should expect when applying the ERTFM in a totally different region?
- Secondly, the ground observations in section 2.5.1 are not adequately introduced. What sort of preprocessing steps were used for quality control? Did you mask out snow and frozen conditions? Did you mask out rainy days which tend create inconsistencies? These have to be very well presented.
- Results: It seems to me that a spatial evaluation is first carried out and then temporal evaluations are also done. For instance, if I am not wrong, Figures 3-11 and Table 1 are all based on spatial evaluations. And then Figures 12-13 deal with temporal evaluations. This must be clarified.
- The Taylor Diagram is quite poorly presented. The green lines in the plot are all 0’s. What does that mean? If they are not necessary, they should be removed. Very little is written on it in the manuscript. Why is it being used? How are you deducing the performance of the models?
- Line 486: “geometry registration”…do you mean georeferencing?
- Line 415: surely, a large epoch for training will result in overfitting...regardless of whatever overfitting method used. A recommended value should be advised for users.
- Line 498: “great superiorty”. This is an overstatement. Please use a better experession.
Reviewer 2 Report
Summary:
In the study titled ‘ERTFM: A Robust Model to Fuse Chinese GF-1 and MODIS Reflectance Data for Terrestrial Latent Heat Flux Estimation’, the authors test a Extremely Randomized Trees Fusion Model (ERTFM) to reconstruct high spatiotemporal resolution reflectance data by fusing the Chinese GaoFen-1 (GF-1) and the Moderate Resolution Imaging 19 Spectroradiometer (MODIS) products. Then they use the new reflectance values and the Modified-Satellite Priestley-Taylor (MS-PT) algorithm to estimate latent heat flux over two eddy covariance tower sites. The model is found to be better than many commonly used fusion techniques. The study is important and would be useful for agricultural monitoring at high spatiotemporal resolution. My main concern is with how the final LE values are evaluated, which could influence the overall results.
Major Comments:
- There is very little information about how the EC data was generated. Are these from both gas analyzer and sonic anemometer observations? What was the surface energy budget closure at this site? Previous studies have shown that the method of closure of the surface energy budget has an impact on evaluations against models (Chakraborty et al. 2019). What happens when a different closure method is used?
- Similarly, the evaluations of LE with the modeled NDVI versus observations is only done for the original MODIS and GF-1 data and then the ERTFM fused product. Why are the other fusion methods not shown here?
- There is an issue with the fused NIR reflectance compared to observations. You can see anomalous values where the observed reflectance does not change much but the predicted values change substantially, consistent for all the methods. Since this affects your NDVI calculations, you need more carefully check why this is happening. Moreover, which pixels on the map do these anomalous values correspond to?
Minor Comments:
1.Figure 3 and Figure 6: Do these use the same range of DN numbers? As in, are the numbers comparable or is this just for visual examination? Either needs a legend for the composite values or more information in the captions.
- Figure 7: What is the red line and what is the black line? Please make the captions more informative.
- Figure 8 and Figure 9 show pretty much similar information. I will either remove one (maybe keep the Taylor diagrams) or move one of them to a supplementary information file.
- Figure 10 and 11: Please change the date format from Julian day to date and month; much easier to understand.
- Figure 12: Subpanel 2 and 4 seem to have 1:1 lines, but 1 and 3 don’t. Please mention what the line colors mean in the captions; same for other similar figures.
- Figure 13: What are the error bars here?
References:
- Chakraborty, T., Sarangi, C., Krishnan, M., Tripathi, S. N., Morrison, R., & Evans, J. (2019). Biases in model-simulated surface energy fluxes during the Indian monsoon onset period. Boundary-Layer Meteorology, 170(2), 323-348.
- Ingwersen, J., Imukova, K., Högy, P., & Streck, T. (2015). On the use of the post-closure methods uncertainty band to evaluate the performance of land surface models against eddy covariance flux data. Biogeosciences, 12(8), 2311-2326.
Round 2
Reviewer 1 Report
I thank the authors for addressing all the comments.
There is just one point that I hope the authors can address in future studies:
The goal of the study was to develop high resolution LE datasets using EFTRM. As mentioned in the first review comments, at such a high resolution, alpha WILL NOT be constant (1.26) for all land cover types. This is the difficulty that comes with high resolution studies. It is not just about increasing pixel resolution, we must also be mindful of the changing physics at such high resolutions. In this study, the focus is on the model development, but this should definitely be followed up with reparameterization of these constant parameters that cannot remain the same when more details at the land surface are being taken into consideration.
All the best!
Reviewer 2 Report
The authors have generally done a good job with the revision. The only confusion I have is regarding Figure 4. The authors mention that they mask out the water pixels. However, this plot shows those errors over water bodies. Are the statistical metrics for all the points shown or only for the non-water pixels? This should be made clear in the captions or the figure should be modified for consistency.
